# Metabolic Adjustments following Glutaminase Inhibition by CB-839 in Glioblastoma Cell Lines

**DOI:** 10.3390/cancers15020531

**Published:** 2023-01-15

**Authors:** Juan De los Santos-Jiménez, Tracy Rosales, Bookyung Ko, José A. Campos-Sandoval, Francisco J. Alonso, Javier Márquez, Ralph J. DeBerardinis, José M. Matés

**Affiliations:** 1Canceromics Laboratory, Departamento de Biología Molecular y Bioquímica, Universidad de Málaga, 29010 Málaga, Spain; 2Instituto de Investigación Biomédica de Málaga (IBIMA-Plataforma BIONAND), Universidad de Málaga, 29010 Málaga, Spain; 3Children’s Medical Center Research Institute, University of Texas Southwestern Medical Center, Dallas, TX 75390, USA; 4Department of Pediatrics, University of Texas Southwestern Medical Center, Dallas, TX 75390, USA; 5McDermott Center for Human Growth and Development, University of Texas Southwestern Medical Center, Dallas, TX 75390, USA; 6Howard Hughes Medical Institute, University of Texas Southwestern Medical Center, Dallas, TX 75390, USA

**Keywords:** aspartate, cancer, cancer metabolism, CB-839, citrate, glioblastoma, glutaminase, glutamine, metabolic reprogramming, metabolomics

## Abstract

**Simple Summary:**

Glioblastoma multiforme is the most common primary brain tumor. Unfortunately, it is also one of the cancer types that has the worst morbidity and mortality ratios, so new targets and treatments need to be found. The metabolism of glutamine is fundamental for the proliferation of many tumor cells, including glioblastomas. Glutaminase isoenzyme GLS is one of the responsible enzymes for the pro-oncogenic pathways that induce metabolic reprogramming and leads to altered levels of some amino acids and other key intermediary metabolites in glioblastoma. Using the clinically approved GLS inhibitor CB-839 (Telaglenastat), we found significant changes in glutamine metabolism, including both the oxidative and reductive fates of Gln-derived alpha-ketoglutarate in the tricarboxylic acid cycle, in three glioblastoma cell lines. One of them, the T98G glioblastoma cell line, showed the greatest modification of metabolite levels involved in the de novo biosynthetic pathways for nucleotides, as well as a higher content of methylated and acetylated metabolites.

**Abstract:**

Most tumor cells can use glutamine (Gln) for energy generation and biosynthetic purposes. Glutaminases (GAs) convert Gln into glutamate and ammonium. In humans, GAs are encoded by two genes: *GLS* and *GLS2*. In glioblastoma, *GLS* is commonly overexpressed and considered pro-oncogenic. We studied the metabolic effects of inhibiting GLS activity in T98G, LN229, and U87MG human glioblastoma cell lines by using the inhibitor CB-839. We performed metabolomics and isotope tracing experiments using U-^13^C-labeled Gln, as well as ^15^N-labeled Gln in the amide group, to determine the metabolic fates of Gln carbon and nitrogen atoms. In the presence of the inhibitor, the results showed an accumulation of Gln and lower levels of tricarboxylic acid cycle intermediates, and aspartate, along with a decreased oxidative labeling and diminished reductive carboxylation-related labeling of these metabolites. Additionally, CB-839 treatment caused decreased levels of metabolites from pyrimidine biosynthesis and an accumulation of intermediate metabolites in the de novo purine nucleotide biosynthesis pathway. The levels of some acetylated and methylated metabolites were significantly increased, including acetyl-carnitine, trimethyl-lysine, and 5-methylcytosine. In conclusion, we analyzed the metabolic landscape caused by the GLS inhibition of CB-839 in human glioma cells, which might lead to the future development of new combination therapies with CB-839.

## 1. Introduction

The reprogramming of cellular metabolism is now considered a hallmark of cancer [1]. The tumor cell needs more efficient energetic pathways compared to normal cells [2]. Consequently, tumor metabolic adaptations are essential to modify, among other aspects, apoptosis and cell proliferation rates. In addition to increased glucose uptake and enhanced glycolytic activity (even in the presence of oxygen), elevated glutamine (Gln) metabolism is a distinctive sign of some cancers, including glioblastoma multiforme [3,4].

Glioblastoma multiforme (GBM), traditionally classified as WHO grade IV astrocytoma, is characterized by high proliferation rates, which are resistant to apoptosis and lead to short patient survival times [5]. Current therapy consists of surgical resection, radiotherapy, and maintenance chemotherapy using the alkylating agent temozolomide (TMZ); however, efficacy remains low, and new therapeutic options are needed [6]. In most cases, GBM arises as primary tumors, comprising a plethora of genetic and epigenetic alterations which frequently include the overamplification and activating of mutations in the epidermal growth factor receptor (*EGFR*), inactivating mutations and/or deletions in the tumor suppressor phosphatase and tensin homolog (*PTEN*) and the cyclin-dependent kinase inhibitor A (*CDKN2A*) genes; all these changes lead to the overactivation of core signaling pathways such as RAS/RAF/MAPK, PI3K/AKT, or cyclin-dependent kinases 4 and 6 (CDK4/6) [7]. 

Unlike primary GBM, the most common alterations in secondary GBM include TP53 and isocitrate dehydrogenase 1/2 (IDH1/2) mutations, wielding fundamental events on epigenetics and cellular metabolism [8]. These genetic alterations could be exploited as targets for GBM therapy [9]. On the other hand, higher glutaminase (GLS) expression in tumors has been correlated with more malignant characteristics, reduced patient survival, and the worst treatment efficacy [10].

Targeting metabolic rewiring is considered a promising strategy for treating cancer and represents a strong means of overcoming resistance to chemotherapeutic agents because of the enormous plasticity of tumor metabolism [2]. Combination therapy consisting of the simultaneous targeting of regulatory nodes remains a capable procedure to improve treatment responses [1]. Among the key regulators of glutaminolysis, oxidative phosphorylation (OXPHOS), and the biosynthesis of nucleotides and amino acids, glutaminase (GA, EC 3.5.1.2) has been investigated as a target in many types of cancers [3]. GA catalyzes the first step of glutaminolysis by transforming Gln into glutamate and ammonium ions in the mitochondria [5]. In humans, GAs are encoded by two genes named *GLS* and *GLS2*. The *GLS* gene codes for two isoforms that arise by alternative splicing, KGA and GAC, which are usually referred to as GLS isoenzymes and are commonly described as having pro-oncogenic properties [3]. *GLS2* encodes LGA and GAB isoforms, whose role in the tumor appears to be more complicated and context-dependent [5]. GBM is among the most aggressive types of cancer and is the most frequently occurring glioma in adults, showing increasing numbers of incidence and mortality [9]. In GBM, GLS is commonly overexpressed and GLS2 is usually silenced, appearing to show some tumor-suppressive properties [11,12]. The overexpression of GLS2 in human glioma cell lines minimizes the malignant phenotype [10], sensitizes cells to oxidative stress by a common mechanism involving the suppression of the PI3K/AKT pathway [13], and induces an antiproliferative response with cell cycle arrest at the G2/M phase and partial translocation of GLS2 proteins to the cell nuclei [14].

Gliomagenesis, GBM invasion, and migration might be addressed by targeting multiple pathways, including glycolysis and apoptosis regulation, through the use of several strategies, i.e., microRNA inhibitors and vaccine immunotherapy [9,15,16]. However, deeper research is necessary to unravel GBM pathophysiology, signaling pathways, and the genetics involved in the proliferation of these tumor cells. A better understanding of the metabolic routes that elicit tumor cell survival and dissemination is the first step to addressing the challenges of GBM treatment. Glutaminolysis is one of the fundamental processes involved in the progression of GBM [17,18,19], but it has not been profusely studied thus far [20]. Recent research using several GBM cell lines has found how the clinically relevant GLS inhibitor CB-839 suppressed GLS enzymatic activity, limiting the influx of Gln derivatives into the tricarboxylic acid cycle (TCAC) [21]. In this work, we studied, in more detail, the metabolic effects of inhibiting GLS activity by using CB-839 in several human glioma cell lines (T98G, LN229, and U87MG) with different Gln and/or GLS dependency. Metabolomics is being used in a high throughput mode to provide a better understanding of metabolism [21]. Potent pipelines for data analysis have been established for metabolomics so that they appear as explanatory methods for general metabolism, studying rewired metabolic circuits following pathological conditions such as cancer, as well as analyzing the metabolic response to specific pharmacological or genetic treatments [22]. Thus, novel results in this field might help precision oncology to further upgrade the diagnosis and treatments for GBM patients and potentially contribute to the development of new combination therapies. 

## 2. Materials and Methods

### 2.1. Chemicals

CB-839 was purchased from Selleckchem (Houston, TX, USA). Dimethyl sulfoxide (DMSO), dimethyl alpha-ketoglutarate (DMKG), methoxyamine hydrochloride, pyridine, and tert-butyldimethylsilyl ether (TBDMS) were obtained from Sigma-Aldrich Co. (St. Louis, MO, USA). U-^13^C-Gln and ^15^N-Gln were obtained from Cambridge Isotope Laboratories (Tewksbury, MA, USA).

### 2.2. Cell Culture and CB-839 Assessment on Cell Proliferation

LN229 cells were cultured in Dulbecco’s modified Eagle medium (DMEM) supplemented with 10% fetal bovine serum (FBS), 100 I.U./mL penicillin, 100 μg/mL streptomycin, and 4 mM L-Gln as described by the authors [11]. T98G and U87MG were cultured in a Minimum Essential Medium (MEM), supplemented with 10% FBS, 100 I.U./mL penicillin, 100 μg/mL streptomycin, and 2 mM L-Gln. The T98G cell line was purchased from the American Type Culture Collection (ATCC, Rockville, MD, USA). LN229 and U87MG cells were kindly provided by Monika Szeliga, Department of Neurotoxicology, Mossakowski Medical Research Centre, Polish Academy of Sciences, Warsaw, Poland. For metabolomics and tracing experiments, cells were treated with 1 µM CB-839 or a vehicle (DMSO) for 24 h. Growth curves were performed using Trypan blue staining and cell counting to ascertain the cytotoxic/cytostatic effects of CB-839 in a dose-dependent manner and employing DMKG, a membrane permeable analog of alpha-ketoglutarate (αKG), as a potential metabolic rescue.

### 2.3. Metabolomics

After treatment with CB-839, the growth medium was discarded, and after that cells were washed with an ice-cold normal saline solution. Following this, 1 mL of cold 80% *v*/*v* methanol was added per dish, the cells were scraped on ice, and the volume was transferred to microfuge tubes and subjected to three freeze–thaw cycles between liquid nitrogen and a 37 °C water bath. Lysates were centrifuged at 4 °C for 15 min at 17,000× *g,* and the metabolite-containing supernatant was transferred to new microfuge tubes and dried overnight in a Speed-Vac concentrator (Thermo Fisher Scientific, SPD1030, Waltham, MA, USA), keeping the tubes open and applying breathable membranes to avoid cross-contamination. Metabolite-containing pellets were redissolved and transferred to LC-MS autosampler vials and injected on a Quadrupole Time of Flight (QTOF) liquid chromatography-mass spectrometry (LC-MS) (Agilent Technologies, 6550 iFunnel, Santa Clara, CA, USA). The total ion count for each sample was employed to normalize metabolite abundances, and a fold-change was calculated by referring experimental normalized abundances to the control ones.

### 2.4. U-^13^C- Glutamine Tracing Experiments

Cells were initially treated with CB-839 1 µM or DMSO for 24 h, then the cells were washed with PBS twice, and a fresh medium containing labeled Gln and the treatment was added to each dish. Cells were traced for 6 h; then ice-cold normal saline was used for washing the cells twice, which were then processed as described before (see Section 2.3), and dried metabolite pellets were incubated with 1% methoxyamine hydrochloride in pyridine for 15 min at 70 °C, following derivatization with TBDMS for 1 h at 70 °C. The volume was transferred to GC-MS autosampler vials, and 1 µL was injected on an Agilent 7890A gas chromatograph coupled to an Agilent 5975C mass selective detector (Agilent Technologies). The results were manually reviewed, and data were expressed as the isotopologue fractions of the total metabolite.

### 2.5. ^15^N-Glutamine Tracing Assays

Cells were traced with media containing treatment and labeled Gln for 6 h, washed with ice-cold saline, and 0.5 mL of cold 80% acetonitrile was added per dish. The samples were processed as described before (see Section 2.3) and analyzed by LC-MS (AB Sciex, QTRAP 6500, Framingham, MA, USA). Isotopic tracing data were expressed as fractions of the total metabolite.

### 2.6. Statistical Analysis

For tracing experiments and CB-839 assessment on cell proliferation, three independent experiments were made in triplicate; metabolomics was made twice in triplicate. Heatmaps showing VIP scores for significantly altered metabolites were generated by using the web tool Metaboanalyst, following PLS-DA analysis. Dot and bar graphs and statistical analysis (Student’s *t*-test for experimental against control samples) were made using GraphPad Prism software. Values were expressed as means ± SD or the mean fold-change. Additionally, Spearman’s rank-order correlation test was performed. *p* < 0.05 was considered statistically significant. Statistical significance was expressed using asterisks, as follows: * *p* < 0.05, ** *p* < 0.01, *** *p* < 0.001, **** *p* < 0.0001.

## 3. Results

### 3.1. Effect of CB-839 on Cell Proliferation in Three Human GBM Cell Lines

The GLS inhibitor CB-839 (1 nM–1 μM) limits the proliferation of T98G, LN229, and U87MG GBM cells in a dose-dependent pattern with variable efficacy. The membrane-permeable analog of αKG (DMKG) was employed as a metabolic rescue. A total of 2 mM DMKG partially rescued cell proliferation in T98G cells treated with CB-839 1 µM, but this rescue effect was smaller in LN229 and U87MG, which was the least affected cell line (Figure 1).

### 3.2. CB-839 Suppresses Both Oxidative Decarboxylation and Reductive Carboxylation of Gln-Derived αKG in the TCAC

Gln is a main donor of carbons into the TCAC through anaplerosis (i.e., glutaminolysis), principally in cancer cells [23]. Gln can feed the TCAC through the generation of glutamate by GAs, and the subsequent formation of αKG, either by transamination or glutamate dehydrogenase (GDH) reactions, which can be metabolized within the TCAC either following oxidative or reductive pathways. To segregate oxidative and reductive TCAC carbon movements, GBM cells were cultured with stable isotopic Gln analogs labeled at the five positions with ^13^C for 6 h. Citrate originated from reductive carboxylation via NADPH-dependent IDH1 and IDH2 and could be distinguished from the TCAC–dependent oxidative decarboxylation by the number of ^13^C nuclei incorporated from [U-^13^C]Gln into citrate (Figure 2).

U-^13^C-Gln tracing experiments showed a consistent decline in relative amounts of m+4 labeled-isotopologues: succinate (m+4), fumarate (m+4), malate (m+4), aspartate (m+4), and citrate (m+4), which indicate the reduced contribution of glutamine oxidation to these metabolites. There was also a reduction in the contribution of reductive-carboxylation-related labeling in the following metabolites: fumarate (m+3), malate (m+3), aspartate (m+3), and citrate (m+5). On average, fractional labeling was reduced five-fold in T98G, which proved to be the most Gln/GA dependent among the GBM cell lines assayed for maintaining TCAC, three-fold in LN229, and two-fold in U87MG, which was found to be the less Gln/GA dependent assayed cell line (Figure 3).

### 3.3. GLS Inhibition by CB-839 Modifies Key Metabolites

For MS-based metabolomics, two to three hundred metabolites were detected in the three assayed GBM cell lines. Hierarchical analysis was carried out, showing significant differential amounts between non-treated and CB-839-treated cell lines, as shown for T98G (Figure 4), LN229 (Figure 5), and U87MG (Figure 6). Of the assayed metabolites, 120-140 were significantly modified following GLS inhibition by CB-839 in T98G and LN229, and 76 metabolites changed significantly in the treated U87MG (Appendix A). Notably, 46 metabolites were significantly altered (increased or decreased) simultaneously in T98G, LN229, and U87MG, while 83 metabolites significantly changed in both T98G and LN229 cell lines (Appendix A). In addition, differences were greater in T98G (Figure 4) and smaller in U87MG (Figure 6). Among the common metabolites, 10 were diminished at least <0.70-fold in all three GBM cell lines: aspartate, alanine, fumarate, malate, o-acetyl-serine, glutamate, αKG, o-succinyl-homoserine, succinate, and citrate (for details see Appendix A). Seven other metabolites were augmented >2.0-fold in all three GBM cell lines: hydroxy-phenyl-lactate, 3,4-dihydroxy-cinnamic acid, pipecolate, 3-methyl-adenine, N-acetyl-serine, betaine, and o-acetyl-carnitine (for details see Appendix A).

### 3.4. CB-839 Reshapes Core Metabolic Pathways in Human GBM Cell Lines

The levels of metabolites associated with the TCAC were strongly decreased following GLS inhibition by CB-839. Fundamental metabolites (i.e., glutamate, αKG, fumarate, malate, aspartate) were sharply diminished in T98G (0.02 to 0.2-fold) and reduced in LN229 (0.1 to 0.5-fold) and U87MG (0.2 to 0.5-fold), as depicted in Figure 7.

Moreover, pyruvate and alanine levels were also depleted upon CB-839 treatment, as shown in Figure 8. Pyruvate levels were significantly diminished in treated T98G and LN229 cells (0.4 to 0.45-fold), while it did not significantly change for the treated U87MG cells; alanine levels were significantly diminished in the three treated cell lines (from 0.05 to 0.35-fold).

Regarding pentose phosphate pathway-related metabolites, we found diminished levels of 6-phosphogluconate and/or 6-phosphoglucono-δ-lactone in CB-839 treated cell lines, as depicted in Figure 9. On the other hand, D-ribose 5-phosphate, fructose 6-phosphate, fructose 1,6-bisphosphate, and sedoheptulose 7-phosphate levels were significantly augmented for the treated T98G and LN229 cells, while U87MG did not show changes for the abundance of these metabolites. 

### 3.5. CB-839 Alters Acetylated Metabolites and Urea Cycle Reactions

The N-acetylation of several amino acids (glutamate, asparagine, serine, glycine) was positively increased following GLS inhibition by CB-839 in the three assayed cell lines (Figure 10). In some cases (e.g., N-acetyl-asparagine), the amounts were correlated with the Gln/GA-dependence for the GBM cell line; thereby, the N-acetyl-asparagine level was augmented 3.4-fold in T98G, 3-fold in LN229, and 1.47-fold in U87MG. Other acetyl-amino acids (e.g., N-acetyl-glutamate) levels were similarly increased in all three cell lines (from 1.5 to 1.9-fold).

A large accumulation (11-fold) was detected for acetyl-carnitine in the treated T98G cells. The levels of acetyl-carnitine doubled for both LN229 and U87MG cell lines treated with CB-839 (Figure 11a). On the other hand, citrulline was significantly augmented following GLS inhibition by CB-839 in all three assayed GBM cell lines (Figure 11b). Conversely, ornithine and putrescine were significantly increased only in the LN229 cell line (Appendix A). In addition, argininosuccinate was significantly decreased in T98G (0.10-fold), LN229 (0.62-fold), and U87MG (0.37-fold), as depicted in Figure 11c.

### 3.6. CB-839 Modifies De Novo Biosynthesis of Nucleotides

The metabolism of purine nucleotides was significantly modified in all GBM cell lines treated with CB-839. Inosine monophosphate (IMP) and the adenosine monophosphate analog AICAR were augmented in the three assayed GBM cell lines (relevant results are depicted in Figure 12a). Again, T98G showed the highest change in purine de novo biosynthesis intermediate levels, rising as much as 16-fold (IMP) and 10-fold (AICAR), while LN229 showed increased IMP (2.17-fold) and U87MG showed increased AICAR (2.65-fold). Uridine monophosphate (UMP) was diminished 0.48-fold and 0.44-fold in T98G and LN229, respectively (Figure 12b). 

Moreover, tracing experiments using Gln labeled with ^15^N in the amide group showed decreased labeling in some purines and pyrimidine metabolites in T98G cells when treated with CB-839. Comparing the control and treated samples, labeling in adenylosuccinate (m+2) was lost, and labeled adenosine monophosphate (AMP) (m+2) dropped from 0.25 to 0.01, while labeled guanosine metabolites (m+3) were also sharply diminished for guanosine (from 0.26 to 0.007), guanosine monophosphate (GMP) (from 0.48 to 0.03), and guanosine diphosphate (GDP) (from 0.53 to 0.04) (Figure 13). As represented in Figure 13, treatment with CB-839 caused the labeling in asparagine (m+1) to decline to 0.5. The results for labeled (m+2) cytidine monophosphate (CMP), cytidine diphosphate (CDP), and cytidine triphosphate (CTP) dropped to very low fractions (0.036, 0.07, and <0.01, respectively). The labeled product (m+1) for uridine monophosphate (UMP), uridine diphosphate (UDP), and uridine triphosphate (UTP) decreased to 0.16, 0.35, and 0.36, respectively.

### 3.7. CB-839 Increases Methylation Profiles

The level of 3-methyl-adenine increased in all the assayed GBM cell lines, as depicted in Figure 14a. CB-839 treatment increased methylated adenine by 2.68-fold for T98G, 2.58-fold for LN229, and 2.47-fold for U87MG. Additionally, the levels of methylated metabolite N,N,N-trimethyl-lysine were significantly increased in all three cell lines analyzed when treated with CB-839. Figure 14b shows a 2.17-fold increase for T98G, 1.4-fold for LN229, and 1.4-fold for U87MG. On the other hand, T98G treated with CB-839 showed a 22-fold increase in 5-methylcytosine levels (Figure 14c).

We wondered if higher levels of methylated metabolites could be attributable to a potential lower activity of αKG-dependent dioxygenases, including histone demethylases and the ten-eleven-translocation (TET) family of 5-methylcytosine hydroxylases. Consequently, we calculated the relative levels of αKG and succinate, the substrate and product, respectively, of αKG-dependent demethylases, and expressed it as the αKG/succinate ratio, which we found to have diminished to 0.13 for T98G, 0.55 for LN229, and 0.55 for U87MG cells, following an inverted pattern compared to the increased levels in methylated metabolites for each treated cell line (Figure 15). 

## 4. Discussion

GLS inhibition by CB-839 induced a dose-dependent decrease in the cell proliferation of all the assayed GBM cell lines (Figure 1), as expected from previous results in several glioma cells [24]. However, significant variation was observed regarding the CB-839 antiproliferative effect among the assayed GBM cell lines, with T98G and LN229 being the most affected ones, while U87MG showed lower affection. A stronger CB-839-dependent antiproliferative effect on cancer cells, compared to non-cancer cells, was determined in the earlier studies, suggesting the use of CB-839 as a promising tool against GBM [6,21]. Interestingly, Poonaki et al. very recently described a method to deliver this drug using gold nanoparticles [24]. The development of this or an alternative nano-carrier to encapsulate the drug might be especially applicable as an adjuvant therapy to enhance the beneficial effect of other treatments in GBM [25]. 

Some tumor cells might use reductive Gln metabolism for the generation of citrate in parallel to the predominant oxidative glutaminolytic pathway [26]. This appears to be the case for the T98G cell line, which we found to have higher reductive (m+5)-associated labeling in citrate from ^13^C-Gln compared to the oxidative (m+4)-related mark (Figure 3a,b). Metabolic reprogramming is a hallmark of cancer [2] but, despite every type of tumor can show different rewiring characteristics to allow for growth and proliferation, Gln addiction is an inherent peculiarity of many cancer cells in culture [8]. Both the decrease in oxidative decarboxylation and reductive carboxylation of Gln-derived αKG in the TCAC, as noted by reduced isotopologue fractions associated with these pathways, are metabolic events following GLS inhibition by CB-839, associated with the change in a myriad of essential metabolites involved in multiple fundamental aspects of cancer. For example, reduced amounts of glutamate, arising from Gln via GA, do not allow the required synthesis of αKG to fuel the high mitochondrial activity through the TCAC for ATP production [3] or their use by chromatin-modifying enzymes, thus impacting proliferation and cell differentiation. Furthermore, the crucial amino acid aspartate is another of the decreased metabolites. Aspartate can be synthesized via aspartate aminotransferase by transferring the amino group from glutamate to oxaloacetate. Hence, lower aspartate levels are probably a consequence of both the lower availability of glutamate and lower levels of TCAC intermediates derived from Gln, including oxaloacetate. Of note, employing DMKG as a metabolic rescue along with CB-839 1 µM partially rescued cell proliferation/cell viability in T98G cells, but showed a smaller rescue effect in the other two cell lines: LN229 and U87MG. These results likely suggest that, in the experimental conditions, GLS significantly contributes to maintaining TCAC intermediate levels, but in many cases, it may not be essential for it. DMKG acts by rescuing αKG levels, which represent the point of entry for glutaminolytic metabolites into the TCAC. However, in this situation, Gln-derived glutamate would be equally depleted, not being able to participate in transamination reactions, which, as discussed before, are essential in the biosynthesis of amino acids such as alanine or aspartate, the latter being indispensable for, among others, the de novo nucleotide biosynthesis pathways. This interpretation is also in line with previous results employing CB-839 in GBM stem cells [18]. T98G cells treated with CB-839 showed the largest reduction in TCAC intermediate levels and labeling from U-^13^C-Gln, which, together with the larger rescue effect of DMKG compared to the other two cell lines assayed, leads to the conclusion that these cells have a high dependency on GLS for maintaining TCAC metabolites levels. It would be of interest to analyze tumor cell GLS dependency for fueling the TCAC as a predictor of CB-839 sensitivity. As discussed before, CB-839 might indirectly target essential metabolite biosynthesis, such as aspartate, in a dual fashion by lowering glutamate, which is needed as an amino group donor, and by lowering glutaminolysis-derived oxaloacetate, acting as an amino acceptor. Bearing this in mind, CB-839’s higher efficacy could be expected in cells with high dependency on GLS activity for maintaining TCAC intermediate levels.

As mentioned before, U87MG cells, when treated with CB-839, showed a much smaller affection to cell proliferation compared to the other two cell lines assayed, as well as overall quantitatively lower changes in metabolomic and isotopic tracing results. Several reasons may be hypothesized to explain this different behavior upon CB-839 treatment, the simpler one being the consideration that U87MG cells are less dependent on Gln, or at least on GLS, for maintaining metabolite levels, especially TCAC-related ones. Of note, untreated U87MG cells showed lower proliferation rates than T98G and LN229, and carbon tracing data showed lower labeled metabolite fractions from Gln in untreated U87MG cells for the same 6 h tracing time compared to the other assayed cell lines, which points to a lower glutaminolytic rate in these cells. It may be speculated that U87MG cells relied more on glucose for maintaining the levels of TCAC metabolites, which could be related to higher pyruvate carboxylase expression for the generation of oxaloacetate from glucose-derived pyruvate, but certainly, further studies are needed to ascertain the molecular explanation behind this differential behavior of cells upon CB-839, and therefore, treatment efficacy.

Gln was shown to sustain citrate synthesis through both glutaminolysis and the reductive carboxylation of αKG in a GLS-dependent manner in prostate organoids, and it was found that Gln could exhibit a stronger contribution to citrate synthesis than glucose and aspartate [27]. These data agree with other studies describing a decoupled TCAC in lung cancer cells, following large increments in glycolysis and Gln reductive carboxylation [22]. Changes in the metabolism of carnitine are markers for gliomas [28,29]. Our results in GBM cell lines agree with this finding, as shown in Figure 11. GLS inhibition by CB-839 rescued the acetyl-carnitine drop by raising more than 11-fold the acetyl-carnitine levels for the most affected cell line, T98G (Figure 11a). We hypothesize that altered acetyl-carnitine levels in this study could be a consequence of increased mitochondrial acetyl-CoA levels. A possible explanation for the hypothetical mitochondrial acetyl-CoA accumulation upon GLS inhibition by CB-839 is based on the depletion of TCAC intermediates, particularly oxaloacetate, which could become limiting for the condensation reaction with acetyl-CoA to form citrate when catalyzed by citrate synthase. Hence, excessive mitochondrial acetyl-CoA could be transferring its acetyl group to carnitine to export it out of the mitochondria while restoring free mitochondrial HS-CoA to be used in various pathways. In fact, acetyl-carnitine is known to be involved in importing acetyl groups into the nucleus, where it can again react with HS-CoA to form acetyl-CoA to participate as an acetyl donor in acetylation reactions [30]. We also hypothesized that increased levels of N-acetyl-amino acids, such as N-acetyl-asparagine, N-acetyl-glycine, N-acetyl-serine, or N-acetyl-glutamate (Figure 10), could indicate a higher availability of acetyl-CoA, which participate in N-acetylation reactions as an acetyl group donor. Noteworthy, N-acetyl-glutamate levels were higher with CB-839 when glutamate levels were much lower, which may be due to acetyl-CoA being the limiting substrate for these N-acetylation reactions. All of that might point to a situation of increased protein acetylation due to the higher availability of acetyl-CoA because of its probable accumulation in the mitochondria.

Equally interesting were the results affecting the adjustment of some of the urea cycle reactions that might justify some fundamental metabolic modifications after GLS inhibition. Citrulline was significantly accumulated in the three assayed cell lines (Figure 11b), which could be explained because of the lower levels of aspartate needed for reacting with citrulline to form argininosuccinate in an ATP-consuming reaction catalyzed by argininosuccinate synthetase. Lower levels of argininosuccinate in the three treated cell lines (Figure 11c) also support this interpretation. There were also some significant changes regarding polyamine levels in the CB-839 treated cell lines, as depicted in Appendix A. However, the expression of the urea cycle enzymes is very variable among tissues, and there is no previous information about its expression in the cell lines employed in this study, so care is to be taken when regarding the interpretation of these metabolic changes. In fact, N-acetyl-glutamate is an essential allosteric activator of carbamoyl phosphate synthetase 1 (CPS1), so it may be hypothesized that if CPS1 is expressed in the analyzed glioma cell lines, its activation by increased levels of N-acetyl-glutamate might also be responsible for the observed citrulline accumulation.

Oxidative stress usually drives pleiotropic metabolic changes [10]. GA inhibition by CB-839 can modify cellular bioenergetics, evoking a reductive change in the NADH/NAD^+^ redox couple [23]. Uridine and aspartate shortage can be a dual effect of both the increased Gln-dependent reductive carboxylation and the variation in redox balance. Pharmacological or genetic silencing of GA in GBM cell lines elicited a clear alteration of redox and bioenergetic pathways [31], including outstanding modifications in reduced glutathione (GSH) levels, GSH/GSSG ratio, and antioxidant enzymes [10]. Our results found diminished amounts of GSH in T98G and LN229 cells after CB-839 treatment (Appendix A). Interestingly, there exists a proven interplay between the TCAC, redox homeostasis, and amino acid metabolism as a consequence of the impairment of OXPHOS following Gln/Glutamate imbalance, leading to lower levels of aspartate, proline, and GSH [32,33].

Gln supplies purine nucleotide and, therefore, nucleic acid synthesis, which is needed to sustain cancer cell proliferation [20]. In the context of highly proliferating cells, very high de novo nucleotide biosynthesis rates are needed for sustaining nucleic acid biosynthesis, given the fact that free nucleotide concentrations in the cell are far below the amounts present in nucleic acids. Our results showed a remarkable accumulation of purine de novo biosynthesis intermediate inosine monophosphate (IMP) and/or AICAR (Figure 12a), depending on the cell line, when treated with CB-839. These changes might be explained because of the need for aspartate, which is depleted in these conditions, to react with IMP to form adenylosuccinate, which can later produce AMP. Although overall AMP levels did not change, the T98G cell line treated with CB-839 showed a reduction in labeling from ^15^N-labeled Gln (amide group) in adenylosuccinate, AMP, and GMP, among others (Figure 13). These results point towards the impairment of the de novo purine biosynthesis pathway by CB-839 treatment, likely attributable to aspartate depletion. In line with these changes, CB-839-treated T98G and LN229 cells showed a significant accumulation of crucial metabolites at the beginning of the de novo purine biosynthesis pathway; ribose 5-phosphate (Figure 9), which serves for synthesizing phosphoribosyl pyrophosphate (PRPP), and glycine (Appendix A), needed to start forming the purine ring by reacting with the phosphoribosyl amine produced after the transfer of the amide group of Gln to PRPP. 6-phosphogluconate or 6-phosphoglucono-δ-lactone as intermediates in ribose 5-phosphate biosynthesis are depleted. In addition, fructose 6-phosphate, fructose 1,6-bisphosphate, and sedoheptulose 7-phosphate, intermediates in the equilibrium reactions of the non-oxidative pentose phosphate pathway, accumulated in the presence of CB-839 (Figure 9). Taken together, these results suggest a lower rate of de novo purine biosynthesis, which again is likely to be caused by aspartate depletion: a key metabolite needed in two different steps of the pathway. On the other hand, pyrimidine de novo biosynthesis is initiated from aspartate. Again, in accordance with the sharp drop in the aspartate level following CB-839 treatment in the assayed GBM cells, the amounts of UMP were strongly decreased (Figure 12b). In line with these results, CB-839-treated T98G cells showed decreased labeling from ^15^N-labeled Gln in the uridine metabolites UMP, UDP, and UTP, as well as in cytidine metabolites CMP, CDP, and CTP (Figure 13e,f). This supports an impaired pyrimidine de novo biosynthesis pathway. Furthermore, CB-839 also caused a reduction in labeled asparagine (Figure 13d), which is synthesized from aspartate and Gln, with the latter acting as an amide group donor.

Of note, CB-839 treatment decreased the succinate biosynthesis derived from glutamine, as noted by the sharp decrease in succinate m+4 labeling for all cell lines. Accordingly, succinate metabolite levels were also down when treated with CB-839. However, as noted by lower αKG/succinate ratios (Figure 15), the drop in succinate levels was quantitatively much smaller compared to upstream and downstream metabolites, such as αKG, fumarate, or malate and also compared to the decrease in succinate labeling from Gln. It is tempting to speculate about the mechanistic reasons causing GLS inhibition by CB-839 to induce such an imbalance in succinate levels compared to the rest of TCAC intermediates since it is unlikely that CB-839 itself has a direct effect on the enzyme metabolizing succinate, succinate dehydrogenase (SDH). One possible hypothesis for explaining that result would be the compensatory activation of a succinate-restoring pathway upon targeting glutaminolysis, such as the degradation pathways of isoleucine, valine, methionine, and threonine. These amino acids can be degraded to provide a final product, succinyl-CoA, in the mitochondria, which can enter the TCAC. In the assayed experimental conditions, these amino acids were supplied by the cell culture media, so it would be reasonable to think of their catabolism to be partially rescuing succinate levels. However, our results show a significant accumulation of methionine, isoleucine, valine, threonine, and also leucine in all or some of the cell lines treated with CB-839, as noted in Appendix A. Accumulation of isoleucine, valine, and leucine may be explained because of the need for αKG for an initial transamination reaction; therefore, depleted αKG caused by CB-839 could be equally limiting the degradation pathways for these metabolites. Hence, CB-839 treatment is apparently preventing these pathways from restoring succinate levels. However, it seems unlikely that a succinate-restoring pathway is the cause of succinate relative accumulation, given the fact that levels of the immediate downstream metabolite from succinate in the TCAC, fumarate, are much lower. So, it seems logical to think that the rate of succinate conversion to fumarate is affected by CB-839, probably in an indirect fashion. A more complex interpretation considers the role of SDH as a common node between the TCAC and the mitochondrial electron transport chain (ETC). SDH is part of Complex II of the ETC and couples with succinate oxidation to fumarate to the concomitant transfer of electrons and protons from succinate to FAD cofactor of SDH to form FADH_2_, which is later used for ubiquinone (Q) reduction to ubiquinol (QH_2_). CB-839 might be indirectly causing a change in the NADH/NAD^+^ ratio while affecting flux through the ETC [34]. Such a landscape could explain the lower succinate oxidation rate to fumarate, given the dependence of this reaction on ubiquinone (oxidized) availability and, hence, undergoes succinate relative accumulation compared to αKG. 

In addition, the results of our study established that GLS inhibition by CB-839 simultaneously provoked TCAC imbalance, as discussed before, causing an asymmetric drop between succinate and other TCAC intermediates, which could originate as a possible malfunction of demethylases. In our models, the levels of the methylated metabolite N,N,N-trimethyl-lysine were significantly increased in all three cell lines analyzed when treated with CB-839. Free N,N,N-trimethyl-lysine is likely to arise from methylated protein degradation, including histones, so it could be interpreted as an indicator of overall protein methylation levels, and this accumulation might also be related to lower levels of L-carnitine found in the three CB-839 treated cell lines (Supplementary Appendix A), due to N,N,N-trimethyl-lysine implication in carnitine biosynthesis pathway [35,36]. Lower levels of L-carnitine in CB-839 treated cells might also be caused by lower levels of αKG, which is needed in the pathway, while glycine accumulation is also in line with these changes. Moreover, T98G treated with CB-839 showed a 22-fold increase in 5-methylcytosine levels, which could equally be attributable to methylated DNA. We hypothesize that superior levels of methylated metabolites could be linked with the lower activity of αKG-dependent dioxygenases, including histone demethylases, prolyl-hydroxylases, and the TET family of 5-methylcytosine hydroxylases, some of which are involved in a multi-step demethylation process of methylated substrates, including DNA or histones (Figure 16). These enzymes use αKG as a substrate and produce succinate. Both lower levels of its substrate αKG and higher proportional succinate, which has been described to compete with αKG for binding to the enzyme causing enzyme inhibition [37], can result in lower dioxygenase activity. In fact, T98G cells treated with CB-839 showed the greatest reduction in αKG/succinate ratio while showing the highest increase in N,N,N-trimethyl-lysine levels, and T98G was the only cell line showing a significant accumulation of 5-methylcytosine. Appropriately, combination therapy targeting both TCAC and DNA demethylases has been proven to provide a synergistic effect against tumor burden in the xenografts of human leukemia cells [38]. Of interest, some essential demethylases have been described as promoting the survival of T98G, LN229, U87MG, and other GBM cell lines by upregulating c-Myc gene expression and inhibiting p53 transcriptional activity [39]. Accordingly, diminished values were found for c-Myc expression when GLS was genetically inhibited in LN229 cells [40]; GLS2 was overexpressed in T98G cells [14]; in the latter model, an increase in p53 levels was also noted [14].

Future studies will allow a better understanding of Gln metabolism in GBM. Indeed, a better comprehension of how GBM cells regulate the TCAC might have potential applications in the use of Gln levels for the imaging analysis of patients suffering from GBM, as well as for their utilization as biomarkers for diagnosis and prognosis [41,42,43]. Additional research focusing on these and other Gln-related issues needs an in-depth exploration and analysis to be developed and applied to clinical practice [44]. Of note, glutaminase inhibitors, particularly CB-839, are being investigated as candidates in several combination therapies [45]. Furthermore, a better understanding of the cell metabolic response upon GLS inhibition by CB-839 is needed. Further studies focusing on potential compensatory pathways, and changes in methylation for both nucleotides and amino acids, might also empower the development of new targeted combination therapies against GBM. It would be of special interest to confirm if these changes in methylation profiles are related to nucleic acids and protein methylation following specific patterns or if they are otherwise unspecific. In this regard, identifying proliferation-limiting pathways that might be synergistically targeted by combination therapies, including CB-839, is significant, as it may be aspartate biosynthesis targeting due to its crucial role in various pathways and might improve the development of new therapeutical strategies. 

## 5. Conclusions

Taken together, our study concluded that a signature metabolic response was demonstrated in GBM cells following specific GLS inhibition by using the allosteric inhibitor CB-839. This mark consists of (i) lowering oxidative and reductive anaplerotic activity, (ii) modifying the de novo synthesis of nucleotides, (iii) increasing acetylation and methylation overall levels, and (iv) reshaping urea cycle reactions. All these findings paved the way for further CB-839 or other GA-targeted studies and extended the basis for its future clinical application in GBM, especially pointing to the future development of combination targeted therapies that are designed based on a specific tumor molecular signature.

## Figures and Tables

**Figure 1 cancers-15-00531-f001:**
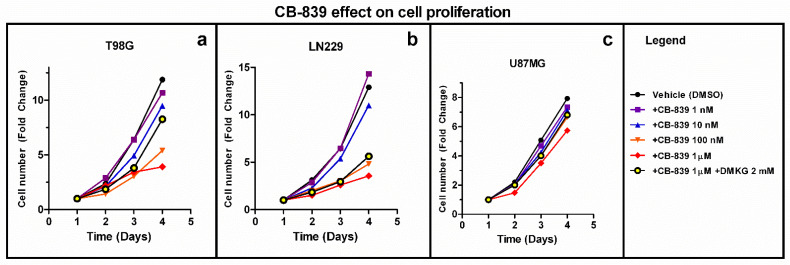
CB-839 inhibits the proliferation of GBM cells in a dose-dependent manner in T98G (**a**), LN229 (**b**) and U87MG (**c**). Dimethyl alpha-ketoglutarate (DMKG) was employed as a metabolic rescue. For each cell line, three independent experiments were made in triplicate. The fold-change in cell number, referring to cells on day 1, is shown.

**Figure 2 cancers-15-00531-f002:**
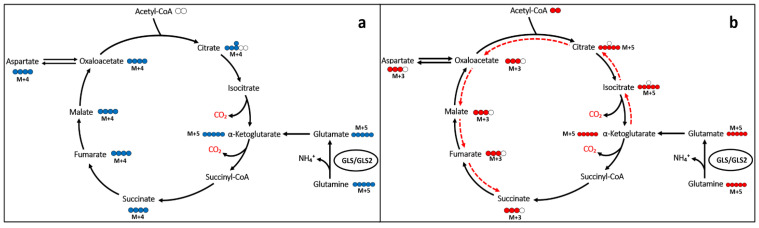
Simplified Gln carbon inheritance scheme (not showing all the reactions, cell compartments, enzymes or metabolites involved). Isotopomers of Gln allowed the specific quantification of Gln carbon contribution to citrate either following an oxidative (**a**) (m+4) or reductive (**b**) (m+5) pathway (labeling for the first turn is shown).

**Figure 3 cancers-15-00531-f003:**
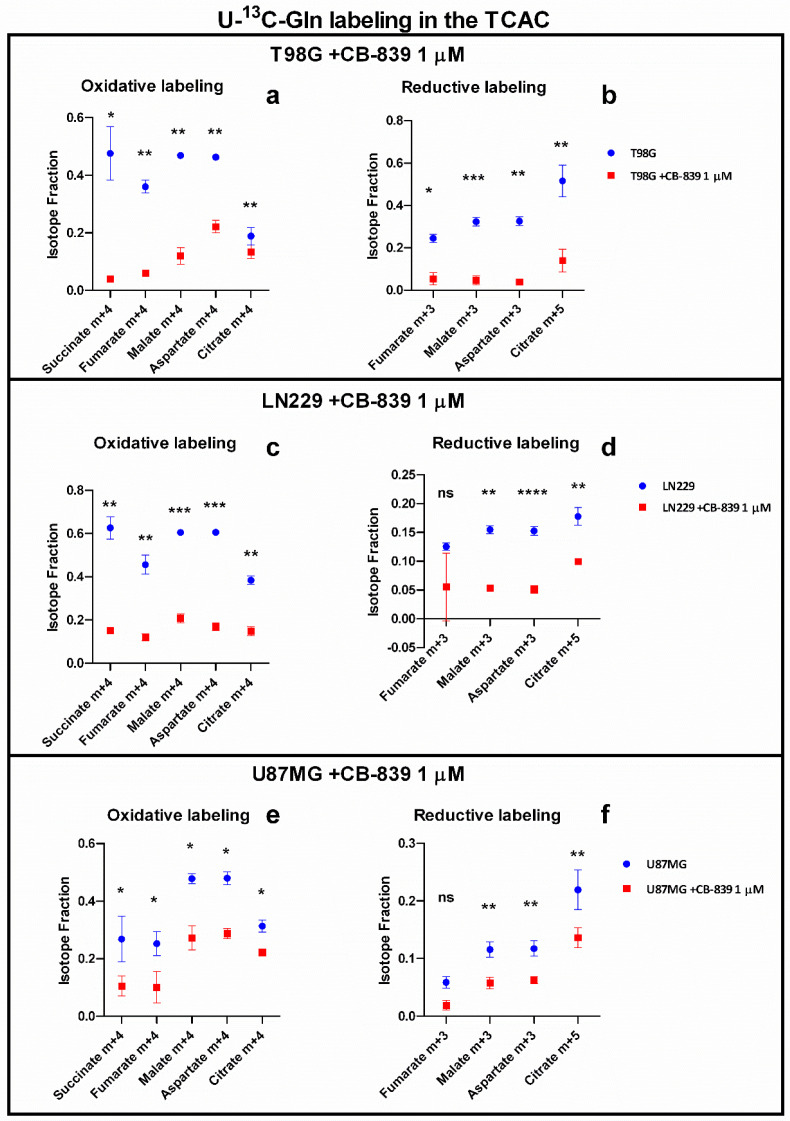
Isotopic tracing in T98G, LN229, and U87MG showing oxidative (**a**,**c**,**e**) and reductive (**b**,**d**,**f**) associated labelling in the TCAC. Results are shown as the isotopologue fraction referred to the total metabolite. The samples were run in triplicate. For more details, see Section 2. ns—non-significant. * *p* < 0.05, ** *p* < 0.01, *** *p* < 0.001, **** *p* < 0.0001.

**Figure 4 cancers-15-00531-f004:**
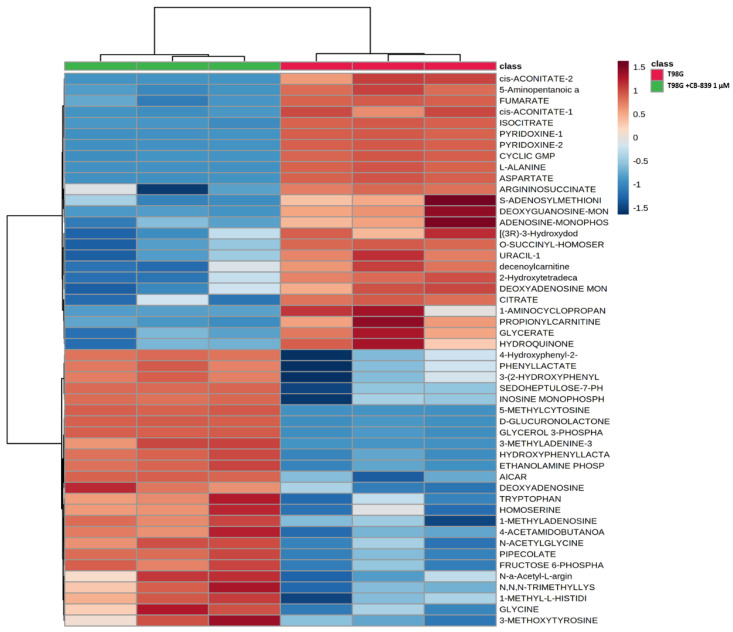
Metabolomic changes elicited by CB-839 are depicted on a representative hierarchical clustering analysis of key molecules in the T98G glioma cell line. The heat map shows the top 50 differentially altered metabolites in GBM cells were exposed for 24 h to 1 μM CB-839. The samples were run in triplicate. For more details, see Section 2.

**Figure 5 cancers-15-00531-f005:**
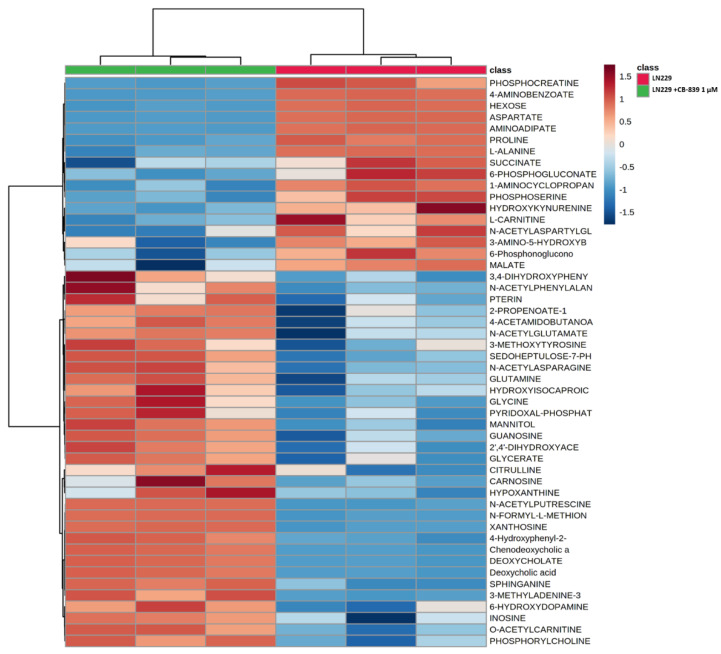
Metabolomic changes elicited by CB-839 are depicted on a representative hierarchical clustering analysis of key molecules in the LN229 glioma cell line. The heat map shows the top 50 differentially altered metabolites in GBM cells exposed for 24 h to 1 μM CB-839. The samples were run in triplicate. For more details, see Section 2.

**Figure 6 cancers-15-00531-f006:**
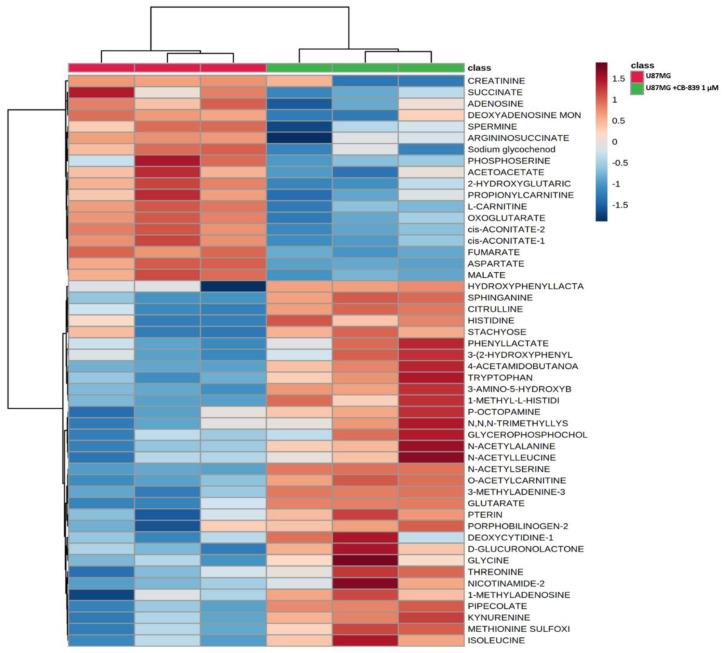
Metabolomic changes elicited by CB-839 are depicted on a representative hierarchical clustering analysis of key molecules in the U87MG glioma cell line. The heat map shows the top 50 differentially altered metabolites in GBM cells exposed for 24 h to 1 μM CB-839. The samples were run in triplicate. For more details, see Section 2.

**Figure 7 cancers-15-00531-f007:**
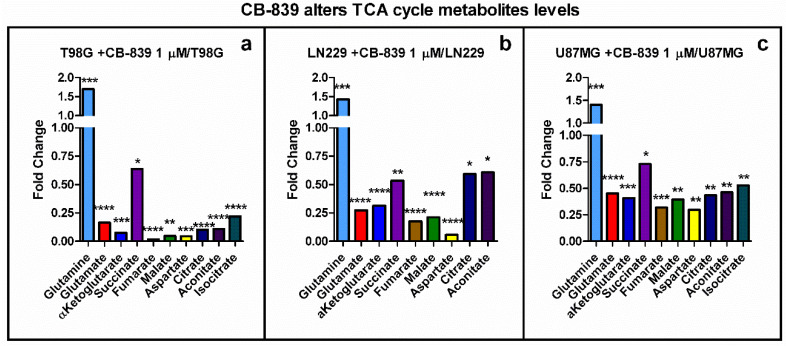
TCAC metabolites altered by specific GLS inhibitor CB-839. Results are depicted for T98G (**a**), LN229 (**b**), and U87MG (**c**). The samples were run in triplicate. For more details, see Section 2. * *p* < 0.05, ** *p* < 0.01, *** *p* < 0.001, **** *p* < 0.0001.

**Figure 8 cancers-15-00531-f008:**
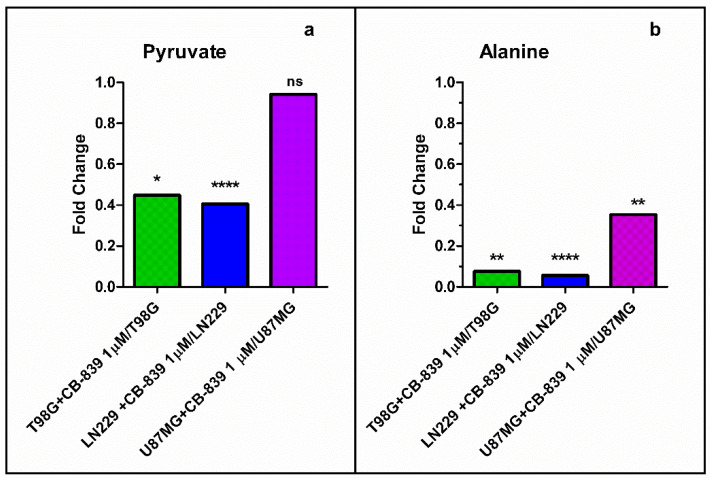
Pyruvate (**a**) and alanine (**b**) levels were diminished upon GLS inhibition by CB-839. The samples were run in triplicate. For more details, see Section 2. ns—non-significant. * *p* < 0.05, ** *p* < 0.01, **** *p* < 0.0001.

**Figure 9 cancers-15-00531-f009:**
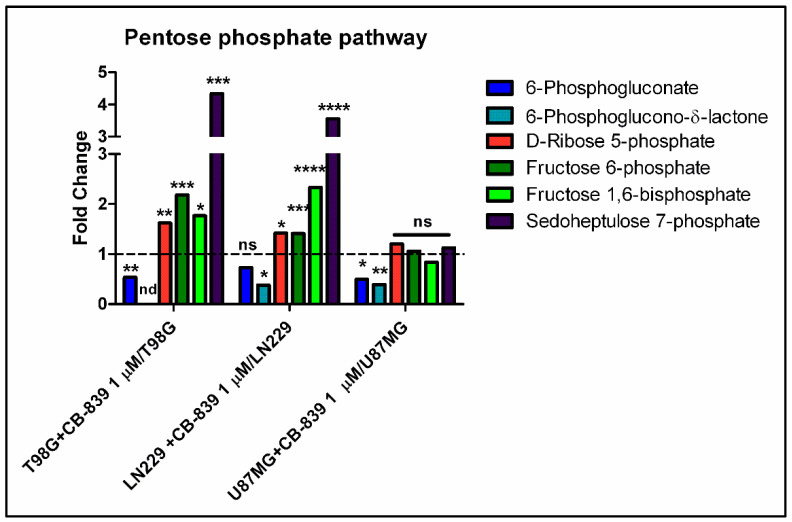
CB-839 altered the abundance of some pentose phosphate pathway-related metabolites. 6-Phosphogluconate was lower in treated T98G and U87MG cells, and 6- phosphoglucono-δ-lactone was diminished in LN229 and U87MG, while it was not detected in T98G. D-ribose 5-phosphate, fructose 6-phosphate, fructose 1,6-bisphosphate, and sedoheptulose 7-phosphate significantly accumulated in CB-839 treated T98G and LN229 cell lines. The samples were run in triplicate. For more details, see Section 2. ns—non-significant. nd—non-detected. * *p* < 0.05, ** *p* < 0.01, *** *p* < 0.001, **** *p* < 0.0001.

**Figure 10 cancers-15-00531-f010:**
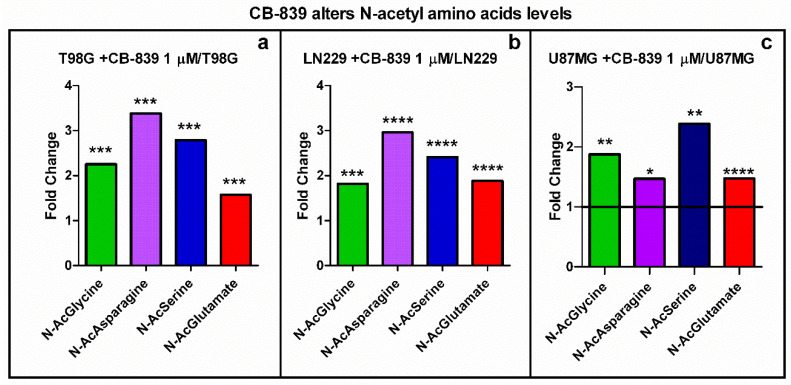
Increased levels of N-acetylated amino acids by specific GLS inhibitor CB-839 in LN229 (**a**), T98G (**b**), and U87MG (**c**). The samples were run in triplicate. For more details, see Section 2. * *p* < 0.05, ** *p* < 0.01, *** *p* < 0.001, **** *p* < 0.0001.

**Figure 11 cancers-15-00531-f011:**
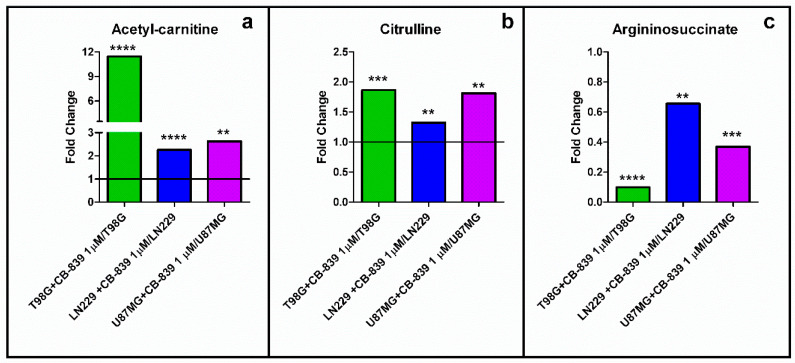
(**a**) Acetyl-carnitine and (**b**) citrulline significantly accumulated, while (**c**) argininosuccinate was depleted, following specific GLS inhibition by CB-839. The samples were run in triplicate. For more details, see Section 2. ** *p* < 0.01, *** *p* < 0.001, **** *p* < 0.0001.

**Figure 12 cancers-15-00531-f012:**
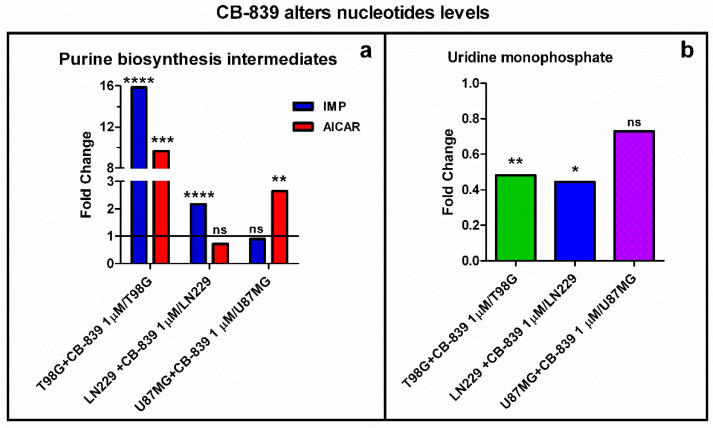
(**a**) Inosine monophosphate (IMP) levels were significantly higher in CB-839 treated T98G and LN229 cells; 5-aminoimidazole-4-carboxamide ribonucleotide (AICAR) levels significantly increased for treated T98G and U87MG cells. (**b**) Uridine monophosphate (UMP) was significantly decreased by specific GLS inhibitor CB-839 in T98G and LN229. The samples were run in triplicate. For more details, see Section 2. ns—non-significant. * *p* < 0.05, ** *p* < 0.01, *** *p* < 0.001, **** *p* < 0.0001.

**Figure 13 cancers-15-00531-f013:**
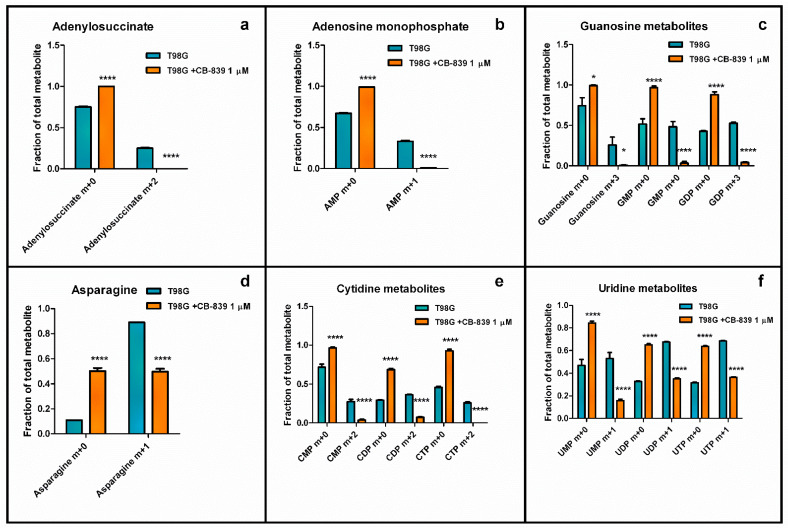
Nitrogen labeling from (amide)-^15^N-Gln in adenylosuccinate (**a**), adenosine monophosphate (AMP) (**b**), guanosine metabolites (**c**), asparagine (**d**), cytidine metabolites (**e**), and uridine metabolites (**f**) were significantly decreased by specific GLS inhibitor CB-839 in human GBM T98G cells. The samples were run in triplicate. For more details, see Section 2. * *p* < 0.05, **** *p* < 0.0001.

**Figure 14 cancers-15-00531-f014:**
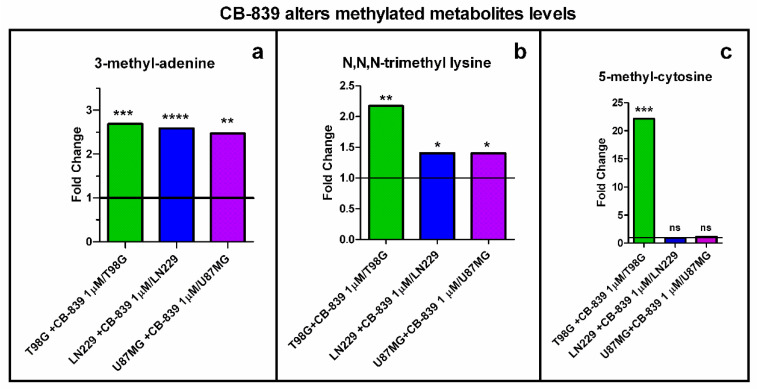
The methylation ratio was significantly increased by treatment with the specific GLS inhibitor CB-839 in the assayed GBM cell lines. Fold change is represented for 3-methyladenine (**a**), N,N,N-trimethyl-lysine (**b**), and 5-methylcytosine (**c**): the last increasing only in T98G. The samples were run in triplicate. For more details, see Section 2. * *p* < 0.05, ** *p* < 0.01, *** *p* < 0.001, **** *p* < 0.0001.

**Figure 15 cancers-15-00531-f015:**
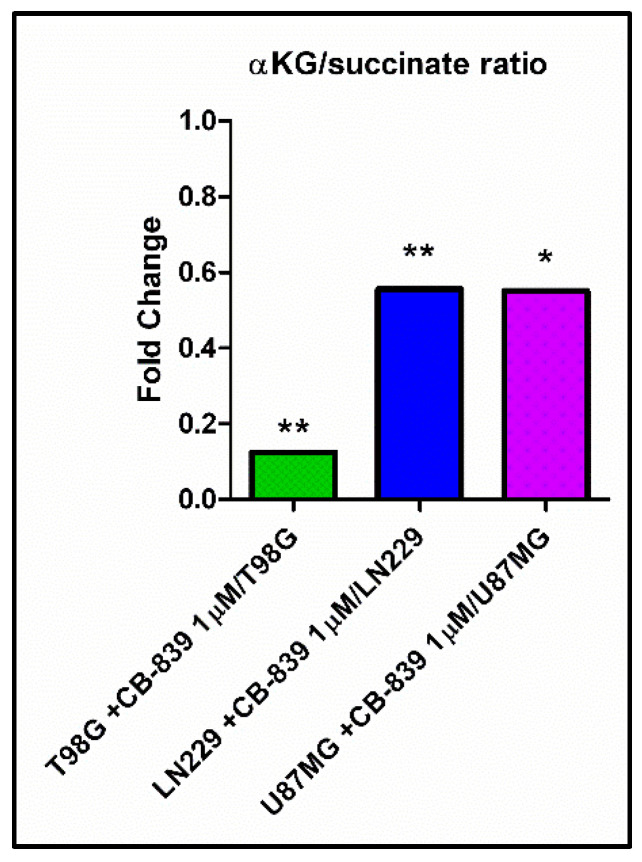
Specific GLS inhibition by CB-839 highly decreased αKG/succinate ratio for all assayed GBM cell lines. The samples were run in triplicate. For further details, see Section 2. * *p* < 0.05, ** *p* < 0.01.

**Figure 16 cancers-15-00531-f016:**
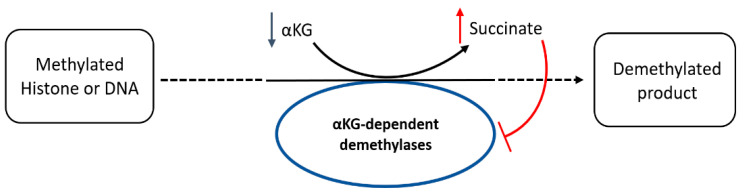
αKG-dependent DNA or protein demethylases use αKG as a substrate and generate succinate as a product, which in high levels can inhibit their activity (oversimplified scheme). A shortage in αKG supply and higher proportional succinate levels can compromise the activity of these key enzymes involved in cell proliferation and differentiation.

## Data Availability

All data generated in this study are available on reasonable request to the corresponding author (J.M.M.).

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
