# Peer review of "Metabolic Adjustments following Glutaminase Inhibition by CB-839 in Glioblastoma Cell Lines"

_cancers, 2023, doi:10.3390/cancers15020531_

Round 1
Reviewer 1 Report
The authors described metabolomic changes by GLS inhibitor CB-839 in three glioblastoma cells and discussed those changes with respect to various metabolic pathways. Overall, the manuscript was written decently and the discussion covered diverse aspects of gln-associated metabolic pathways. The results are interesting enough to draw attention from those who study metabolomics of glioblastoma as well as of other cancers. Here are a few minor concerns on the manuscript.
1. Throughout the manuscript, U87MG manifests quite different CB-839-induced proliferation pattern and metabolomic profiles from the other two cells. It is advised to discuss in detail what could make differences in proliferation and other metabolic changes in response to CB-839 among the three cells, especially focusing on U87MG.
2. Was the rescuing effect of DMKG on metabolic changes, other than proliferation pattern elicited by CB-839 analyzed? If any, show the results.
3. In Fig. 17c shows only T98G result. Is there any reason to show the result in that way?
4. Title is missing in reference 6.
Author Response
- Throughout the manuscript, U87MG manifests quite different CB-839-induced proliferation pattern and metabolomic profiles from the other two cells. It is advised to discuss in detail what could make differences in proliferation and other metabolic changes in response to CB-839 among the three cells, especially focusing on U87MG.
Thank you, we do agree about the value of comparing U87MG, a probable CB-839-treatment “non-responding” cell line to the potential “responding” T98G and LN229 cell lines. We included an additional section in the discussion in that regard. However, as we discussed in the manuscript, further studies are clearly needed to fully uncover the metabolic profile and other molecular characteristics making a tumor cell vulnerable or not to GLS inhibition.
- Was the rescuing effect of DMKG on metabolic changes, other than proliferation pattern elicited by CB-839 analyzed? If any, show the results.
Unfortunately, it was not possible for us to include this rescue condition using DMKG neither in the metabolomic nor isotope tracing studies. It certainly would be of great interest to analyze this condition in the future, since it would make it possible to differentiate CB-839 effects due to the affection of routes in which glutamine-derived alpha-ketoglutarate is implied, like the TCA cycle or the potential changes in methylation that we describe and hypothesize could be related to lower alpha-ketoglutarate-dependent dioxygenases activity.
- In Fig. 17c shows only T98G result. Is there any reason to show the result in that way?
We initially designed the figure that way because T98G was the only cell line showing significant changes in 5-methyl-cytosine levels. However, we agree with the reviewer that it is more appropriate to show all results, which is also in line with other figures in the manuscript. In accordance, we modified the figure to also include LN229 and U87MG results.
- Title is missing in reference 6.
Thank you. Reference 6 has been corrected.
Please, see in the attachment the corrected manuscript showing all the changes suggested by reviewers.

Reviewer 2 Report
The Manuscript by J. de los Santos-Jiménez, T. Rosales, B. Ko, J.A. Campos-Sandoval, F.J. Alonso, J. Márquez, R.J. DeBerardinis, J.M. Matés “Metabolic Adjustment Following Glutaminase Inhibition by CB-839 in Glioblastoma Cell Lines” describes the study of the action of glutaminase inhibitor CB-839 on three cell lines of glioblastoma (U87 MG, LN229 and T98G). A particular attention is paid to the influence of glutaminase inhibition on the various metabolic pathways. Metabolomic and isotope tracing experiments clearly demonstrates that inhibition of single enzyme, glutaminase leads to multiple inracellular effects (e.g., decreasing the level of α-ketogluratate which affects tricarboxylic acid cycle, decreasing the level of aspartate that affects urea cycle reactions, increasing the levels of acetylation and methylation). The diversity of glutaminase inhibition effects allows better understanding the interdependence between key metabolites.
The manuscript is very valuable from the standpoint of biochemistry and molecular biology, because it illustrates both the relationships of metabolic pathways and possibility to observe metabolic effects by physicochemical methods. The research methodology reveals the complex action of glutaminase inhibition by CB-839 (Telaglenastat) which does limit the proliferation of glioblastoma cells, especially T98G.
I believe that the article by J. de los Santos-Jiménez and co-authors will be interesting for the broad auditory of the Cancers journal and recommend to publish it in the present form.
Author Response
Thank you very much, we do appreciate your valuable considerations about our work.

Reviewer 3 Report
CB-839 is a glutaminase inhibitor researched for several types of cancer treatment. The authors in this manuscript found CB-839 inhibits glioblastoma cell proliferation in a dose-dependent manner. They measured the metabolic changes among three glioblastoma cell lines with or without CB-839 treatment. The result clearly shows that several metabolites significantly changed, including TCA metabolites, nucleotides, and amino acids. Overall, this manuscript is enough to be published.
Questions:
1. In Figure 1, if you did three independent experiments, please give all data (with an error bar).
2. Give a summary (make a table, you can just list it in supplementary files) to show the metabolic difference between T98G and U87MG cell lines. CB-839 inhibition and DMKG have different patterns in these two cell lines, which means CB-839 specifically inhibits certain types of cancer cells. The full picture of metabolic differences may give more clues about it.
3. Figure number is too much. You can put some figures in one Figure, such as you can put Figure 3, Figure 4, and Figure 5 in one Figure, and labeled them as “a, b, c, d, e, and f”.
Author Response
CB-839 is a glutaminase inhibitor researched for several types of cancer treatment. The authors in this manuscript found CB-839 inhibits glioblastoma cell proliferation in a dose-dependent manner. They measured the metabolic changes among three glioblastoma cell lines with or without CB-839 treatment. The result clearly shows that several metabolites significantly changed, including TCA metabolites, nucleotides, and amino acids. Overall, this manuscript is enough to be published.
Questions:
- In Figure 1, if you did three independent experiments, please give all data (with an error bar).
We did three independent experiments, however, we decided to represent the mean values of each condition without the error bars, looking for a clearer visual interpretation of the results: since we included a range of CB-839 dosage, error bars of close concentrations (e.g. 1 nM and 10 nM) overlap, and make it difficult to visualize the dots in each condition, so we think make it difficult to clearly interprete the results. Please, see in the attachment the figure version using error bars corresponding to the SEM.
- Give a summary (make a table, you can just list it in supplementary files) to show the metabolic difference between T98G and U87MG cell lines. CB-839 inhibition and DMKG have different patterns in these two cell lines, which means CB-839 specifically inhibits certain types of cancer cells. The full picture of metabolic differences may give more clues about it.
Thank you for your suggestion. We included an additional table in the supplementary files (see Supplementary Table 4) including all the metabolites that significantly change (fold-changes are shown) in the three treated cell lines, so it may serve for an easier comparison of the results among the three assayed cell lines.
- Figure number is too much. You can put some figures in one Figure, such as you can put Figure 3, Figure 4, and Figure 5 in one Figure, and labeled them as “a, b, c, d, e, and f”.
Thank you, we do agree with your consideration. We put together figures 3, 4 and 5 as suggested, and also figures 15 and 16, so we reduced figure number in the manuscript.
